# Prognostic value of SUVmax on 18F-fluorodeoxyglucose PET/CT scan in patients with malignant pleural mesothelioma

**Jun Hyeok Lim[1,2], Joon Young Choi[3], Yunjoo Im[1], Hongseok Yoo[1], Byung Woo Jhun[1], Byeong-Ho Jeong****[1], Hye Yun Park[1], Kyungjong Lee[1], Hojoong Kim[1], O Jung Kwon[1], Joungho Han[4], Myung-Ju Ahn[5], Jhingook Kim[6], Sang-Won Um****[1]\***

**1** Division of Pulmonary and Critical Care Medicine, Department of Medicine, Samsung Medical Center, Sungkyunkwan University School of Medicine, Seoul, South Korea, **2** Division of Pulmonology, Department of Internal Medicine, Inha University Hospital, Inha University School of Medicine, Incheon, South Korea, **3** Department of Nuclear Medicine, Samsung Medical Center, Sungkyunkwan University School of Medicine, Seoul, South Korea, **4** Department of Pathology, Samsung Medical Center, Sungkyunkwan University School of Medicine, Seoul, South Korea, **5** Division of Hematology-Oncology, Department of Medicine, Samsung Medical Center, Sungkyunkwan University School of Medicine, Seoul, South Korea, **6** Department of Thoracic and Cardiovascular Surgery, Samsung Medical Center, Sungkyunkwan University School of Medicine, Seoul, South Korea

\* sangwonum@skku.edu

**Data Availability Statement:** All relevant data are within the paper.

**Funding:** S-W.U. was supported by a grant of the Korea Health Technology R&D Project through the

## Abstract

### Introduction

The maximum standardized uptake value (SUVmax) in $^{18}$F-fluorodeoxyglucose positron emission tomography/computed tomography (PET/CT) may be of prognostic significance for patients with malignant pleural mesothelioma (MPM). This retrospective study aimed to investigate the prognostic value of the SUVmax in patients with MPM.

### Materials and methods

Medical records were retrospectively reviewed for the patients who were diagnosed with histopathologically proven MPM between 2009 and 2018 at Samsung Medical Center. For each patient, SUVmax was calculated for the primary lesion on PET/CT. To determine optimal cutoff values for predicting mortality, receiver operating characteristic curves were used.

### Results

Among the 54 study patients, 34 (63.0%) had epithelioid subtype, 13 (24.1%) had sarcomatoid or biphasic subtype, and 7 (13.0%) had mesothelioma, not otherwise specified (NOS). The median overall survival (OS) was 8.7 months, and the median SUVmax was 9.9. The median values of SUVmax were 5.5 in patients with epithelioid subtype, 11.7 in those with sarcomatoid/biphasic subtype, and 13.3 in those with NOS subtype ($P = 0.003$). The optimal cutoff values of SUVmax to predict mortality were 10.1 in all patients, and 8.5 in patients with epithelioid subtype. In multivariate analysis, SUVmax was significantly associated with overall survival in all patients ($P = 0.003$) and in patients with epithelioid subtype ($P = 0.012$), but not in those with non-epithelioid subtype.

Korea Health Industry Development Institute (KHIDI), funded by the Ministry of Health & Welfare, Korea (HI14C3418). The funders had no role in study design, data collection and analysis, decision to publish, or preparation of the manuscript.

**Competing interests:** The authors have declared that no competing interests exist.

## Conclusions

SUVmax in PET/CT is an independent prognostic factor in patients with MPM, especially those with epithelioid subtype. The histologic subtype of MPM should be considered when evaluating the prognostic significance of SUVmax.

## Introduction

Malignant pleural mesothelioma (MPM) is a rare but aggressive tumor that arises from pleural mesothelial cells. The prognosis of patients with MPM is poor, with a median survival of 20–29 months despite tri-modality treatment including surgery, chemotherapy, and radiotherapy [1, 2]. Surgical methods (e.g., extra-pleural pneumonectomy [EPP] or pleurectomy/decortication) should be selected in accordance with the patient's condition [3, 4]. Among chemotherapeutic agents, a pemetrexed and platinum-based regimen has been recommended as a first-line treatment because of its proven ability to improve the survival rate [5, 6]. Immune checkpoint inhibitors, vinorelbine and gemcitabine are recommended as subsequent systemic therapy in the most recent guideline [6]. Pembrolizumab or nivolumab with (or without) ipilimumab showed promising results in recent clinical trials [7–9].

Predicting the prognosis of patients with MPM is important for determining treatment options. There are multiple prognostic prediction models for MPM, such as the model developed by the European Organization for the Research and Treatment of Cancer (EORTC) and that developed by Cancer and Leukemia Group B (CALGB) [10, 11]. Several studies have reported that $^{18}$F-fluorodeoxyglucose ($^{18}$F-FDG) positron emission tomography (PET) parameters, including maximum standardized uptake value (SUVmax), are associated with the prognosis of MPM [12–19]. Few studies have considered clinical factors such as stage, histology, or chemotherapeutic agents as confounding factors in determining the prognosis of patients with MPM. Because most previous studies are based on PET rather than integrated PET/computed tomography (PET/CT), the applications of the results of these studies in the medical field are limited.

The purpose of this study was to investigate the prognostic value of SUVmax of $^{18}$F-FDG PET/CT in patients with MPM and to define its impact on survival prognosis in those patients. The prognostic value of SUVmax was evaluated for each subgroup based on clinical characteristics.

## Materials and methods

### Patients

We conducted a retrospective review of the medical records of 123 patients who were diagnosed with histopathologically proven MPM during the period between January 2009 and June 2018 at Samsung Medical Center in Seoul, South Korea. In all patients, surgical biopsy was performed for diagnosis of MPM. Patients who were lost to follow-up (n = 4), who did not undergo $^{18}$F-FDG PET/CT (n = 49), or who had no available data for SUV (n = 16) were excluded. Ultimately, 54 patients were enrolled in this retrospective study (Fig 1).

We reviewed clinical records for age, gender, smoking history, exposure to asbestos, location of tumor, presence of bilateral pleural plaque, histologic subtype, stage, SUVmax, type of surgery, and chemotherapy. All patients underwent diagnostic contrast-enhanced CT of the chest and abdomen and $^{18}$F-FDG PET/CT. Disease stage was classified in accordance with the

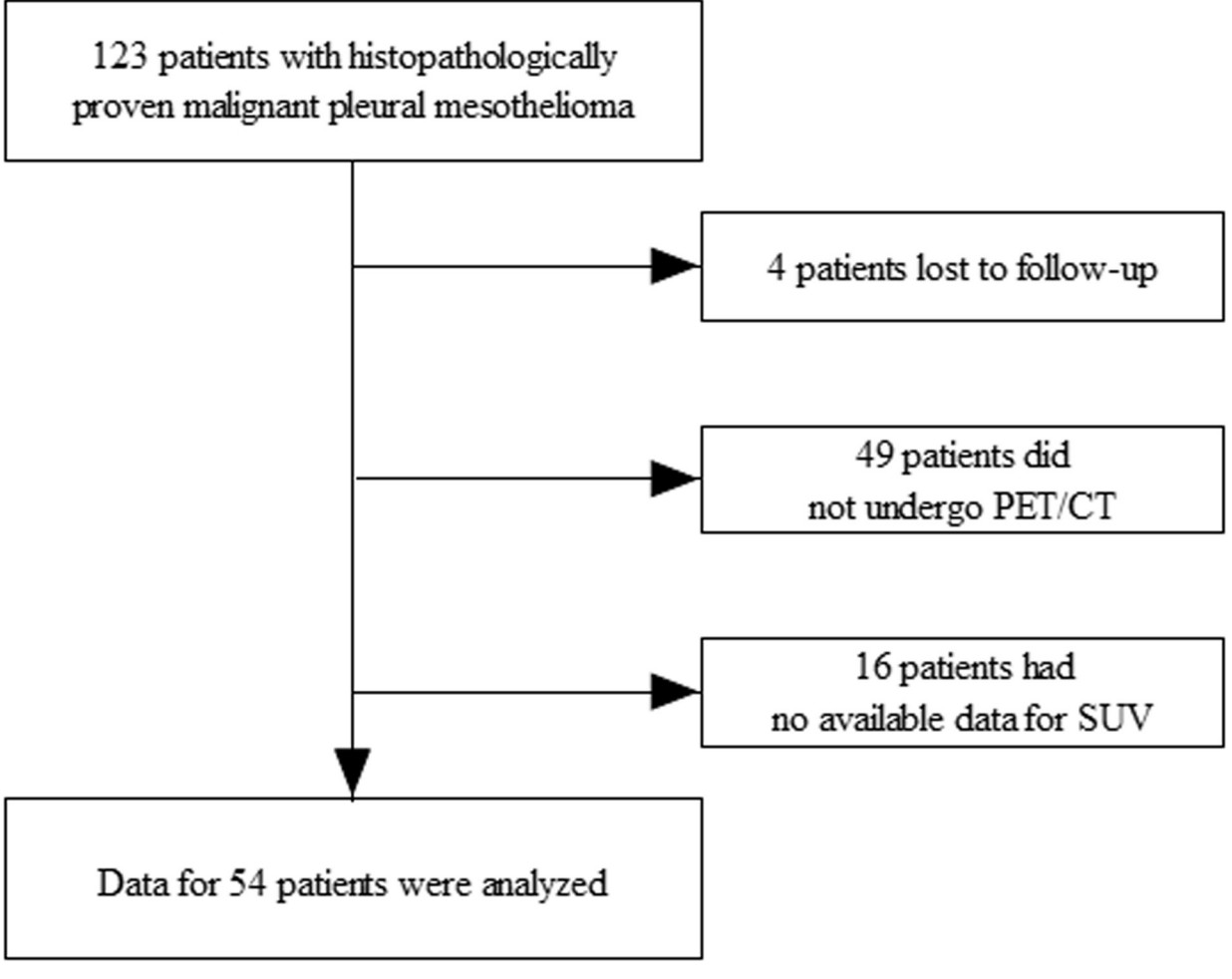

**Fig 1. Flow chart of patients in the study.**

eighth edition of the tumor-node-metastasis (TNM) classification for MPM by the Union for International Cancer Control (UICC) and the American Joint Commission on Cancer (AJCC) [20].

EPP, pleurectomy/decortication, or partial pleurectomy was performed in patients with resectable MPM who could tolerate aggressive surgery. Neoadjuvant or adjuvant chemotherapy with four to six cycles of pemetrexed and cisplatin or carboplatin was administered in combination with surgery. In patients who were not candidates for surgery, palliative chemotherapy was administered with pemetrexed and cisplatin or carboplatin. Cycles of chemotherapy were repeated at 21-day intervals.

This review was approved by the Institutional Review Board of Samsung Medical Center (IRB No. 2018-07-081), which waived the requirement for informed consent by individual patients because of the retrospective nature of the study.

## FDG PET/CT

$^{18}$F-FDG PET/CT was performed prior to surgery or chemotherapy for baseline analysis in all patients. All patients fasted for at least 6 h and had a blood glucose level <150 mg at the time of PET/CT. Imaging was performed 60 min after injection of 5 MBq/kg $^{18}$F-FDG (without

intravenous or oral contrast) on a Discovery LS (GE Healthcare, Waukesha, WI, USA) or a Discovery STe PET/CT scanner (GE Healthcare Waukesha, WI, USA). Continuous spiral CT was performed using an 8-slice helical CT (140 keV; 40–120 mA; Discovery LS) or with 16-slice helical CT (140 keV; 30–170 mA; Discovery STe). Further details were described in our previous published study [21].

The $^{18}$F-FDG PET/CT data were evaluated using the SUVmax by one experienced nuclear medicine physician (J.Y.C) who was blinded to patient outcome. Region of interest analysis tools included with the scanner were used to calculate the SUVmax over the primary tumor after correction for the injected dose of $^{18}$F-FDG and patient weight.

## Statistical analysis

The data are presented as number (%) or median (interquartile range) unless otherwise stated. To compare SUVmax according to clinical characteristics, we performed independent sample *t*-tests or Mann–Whitney *U* tests. Receiver operating characteristic (ROC) curves were plotted to determine the optimal cutoff values of SUVmax that yielded the maximal sensitivity plus specificity of predicting the overall survival. The patient population was subdivided using the cutoff values of SUVmax from the ROC curves, and the duration of overall survival was compared between groups. Overall survival (OS) was calculated as the time (months) from diagnosis until death from any cause. Patients who were alive on the date of the most recent follow-up were censored on that date. Median OS was calculated using the Kaplan–Meier method and compared using a log-rank test. To assess the potential independent effects of SUVmax on OS, we performed univariate and multivariate analyses using Cox proportional hazards models with variables that had *P*-values <0.05. Statistical analyses were performed using a statistical software package (SPSS version 19.0, SPSS, Chicago, IL, USA).

## Results

### Patients

The characteristics of the 54 study patients are summarized in Table 1. The median age was 64 years and 75.9% of patients were men. Thirty-four patients (63.0%) had epithelioid subtype, 13 patients (24.1%) had sarcomatoid (n = 10) or biphasic (n = 3) subtype, and 7 patients (13.0%) had mesothelioma, not otherwise specified (NOS). Nineteen patients (35.2%) underwent surgical resection (EPP [n = 10], pleurectomy/decortication [n = 4] or pleurectomy alone without decortication [n = 5]). Thirty-six patients (66.7% underwent chemotherapy with pemetrexed plus cisplatin or carboplatin (neoadjuvant or adjuvant [n = 11] or palliative chemotherapy [n = 25]). During a median follow-up of 8.7 months (3.8–21.9 months), 30 of 54 (55.6%) MPM patients died. The median OS of patients was 12.6 months.

### SUVmax according to clinical characteristics

The median value of SUVmax was significantly lower in patients with epithelioid subtype (5.5) than in those with sarcomatoid/biphasic subtype (11.7) or mesothelioma, NOS (13.3) (Table 2). The SUVmax was also significantly associated with stage and surgery. The ROC curve showed that the optimal cutoff value of SUVmax for predicting death was 10.1 (area under the curve [AUC] = 0.681) in all patients. Because there was a significant difference in the median SUVmax in relation to tumor subtype, we calculated the optimal cutoff values of SUVmax in relation to tumor subtype. In patients with epithelioid subtype (n = 34), the optimal cutoff value of SUVmax for predicting death was 8.5 (AUC = 0.611). In patients with non-

**Table 1. Baseline characteristics of study subjects.**

| Characteristics | N (%) or Median (IQR) |
|---|---|
| Age (years) | 64 (53–71) |
| Male/female | 41 (75.9)/13 (24.1) |
| Smoker/nonsmoker | 30 (55.6)/24 (44.4) |
| Asbestos exposure | |
| Yes | 15 (27.8) |
| No | 20 (37.0) |
| Unknown | 19 (35.2) |
| Location of tumor | |
| Right | 31 (57.4) |
| Left | 23 (42.6) |
| Bilateral pleural plaque | |
| Yes | 10 (18.5) |
| No | 44 (81.5) |
| Histologic subtype | |
| Epithelioid | 34 (63.0) |
| Sarcomatoid | 10 (18.5) |
| Biphasic | 3 (5.5) |
| NOS | 7 (13.0) |
| T stage | |
| T1 | 15 (27.8) |
| T2 | 6 (11.1) |
| T3 | 14 (25.9) |
| T4 | 19 (35.2) |
| N stage | |
| N0 | 27 (50.0) |
| N1 | 16 (29.6) |
| N2 | 11 (20.4) |
| M stage | |
| M0 | 42 (77.8) |
| M1 | 12 (22.2) |
| Stage | |
| IA | 3 (5.6) |
| IB | 15 (27.8) |
| II | 2 (3.7) |
| IIIA | 4 (7.4) |
| IIIB | 20 (37.0) |
| IV | 10 (18.5) |
| SUVmax | 9.9 (4.4–13.5) |
| Type of surgery | |
| Extrapleural pneumonectomy | 10 (18.5) |
| Pleurectomy/decortication | 4 (7.4) |
| Partial pleurectomy | 5 (9.3) |
| None | 35 (64.8) |
| Chemotherapy | |
| Pemetrexed/platinum | 36 (66.7) |
| None | 18 (33.3) |

IQR, interquartile range; NOS, not otherwise specified; SUVmax, maximum standardized uptake value

**Table 2. Comparison of SUVmax according to clinical characteristics.**

|  | SUVmax | P |
|---|---|---|
| Gender |  | 0.424 |
| Male ($n$ = 41) | 9.7 (3.5–13.5) |  |
| Female ($n$ = 13) | 10.1 (7.0–13.5) |  |
| Histologic subtype |  | 0.003 |
| Epithelioid ($n$ = 34) | 5.5 (3.2–10.8) |  |
| Sarcomatoid/biphasic ($n$ = 13) | 11.7 (9.9–14.7) |  |
| NOS ($n$ = 7) | 13.3 (9.5–15.8) |  |
| Stage |  | 0.031 |
| Stage I–II ($n$ = 20) | 5.5 (3.4–10.8) |  |
| Stage III–IV ($n$ = 34) | 10.4 (7.3–13.7) |  |
| Surgery |  | 0.037 |
| Yes ($n$ = 19) | 5.1 (3.0–10.4) |  |
| No ($n$ = 35) | 10.3 (5.8–13.7) |  |
| Chemotherapy |  | 0.565 |
| Pemetrexed/platinum ($n$ = 36) | 9.1 (4.3–13.5) |  |
| None ($n$ = 18) | 10.3 (4.2–13.3) |  |

Data are presented as median (interquartile range).

NOS, not otherwise specified; SUVmax, maximum standardized uptake value

epithelioid subtype (n = 20) including sarcomatoid/biphasic subtype and mesothelioma, NOS, the optimal cutoff value of SUVmax was 10.3 (AUC = 0.453).

## Univariate survival analysis in relation to clinical characteristics

Univariate analysis of OS included age, gender, smoking history, exposure to asbestos, tumor location, histologic subtype, stage, SUVmax, EPP, and chemotherapy (Table 3). Among all patients, histologic subtype ($P$ < 0.001) (Fig 2A), stage ($P$ = 0.001) (Fig 2B and 2C), SUVmax ($P$ < 0.001) (Fig 2D), and chemotherapy ($P$ = 0.031) were significantly associated with OS. In patients with epithelioid subtype, stage ($P$ = 0.013) and SUVmax ($P$ = 0.007) (Fig 2E) were associated with OS. However, in patients with non-epithelioid subtype, chemotherapy was associated with OS ($P$ = 0.005) but SUVmax was not associated with OS ($P$ = 0.266) (Fig 2F).

## Multivariate survival analysis

SUVmax, subtype, stage, and chemotherapy were included in multivariate analysis (Table 4). SUVmax ($P$ = 0.003), histologic subtype ($P$ = 0.003), stage ($P$ = 0.001), and chemotherapy ($P$ = 0.015) remained significant in all patients. Furthermore, SUVmax ($P$ = 0.012) and stage ($P$ = 0.014) remained significant in patients with epithelioid subtype. In patients with non-epithelioid subtype, chemotherapy ($P$ = 0.044) showed significance in multivariate analysis.

## Discussion

In the present study, we confirmed that SUVmax in PET/CT was an independent prognostic factor for OS in multivariate analysis. Furthermore, subgroup analysis revealed that the SUVmax was a prognostic factor in patients with epithelioid subtype, but not in those with non-epithelioid subtype. Previous studies suggested a relationship between SUVmax and OS in MPM patients [12–19]. However, this was the first study to suggest that the prognostic role of SUVmax could be limited to the epithelioid subtype only. Histologic subtype, stage, and

**Table 3. Univariate analysis for overall survival.**

| | Total (n = 54) | | | Epithelioid (n = 34) | | | Non-epithelioid (n = 20) | | |
|---|---|---|---|---|---|---|---|---|---|
| | Median Survival (months) | 1-year Survival (%) | Log-rank P | Median Survival (months) | 1-year Survival (%) | Log-rank P | Median Survival (months) | 1-year Survival (%) | Log-rank P |
| Age | | | 0.269 | | | 0.367 | | | 0.307 |
| >64 | 11.4 | 30.8 | | 22.5 | 41.2 | | 4.2 | 11.1 | |
| ≤64 | 17.2 | 53.6 | | NR | 76.5 | | 7.1 | 18.2 | |
| Gender | | | 0.558 | | | 0.923 | | | 0.750 |
| Male | 15.3 | 41.5 | | 37.8 | 55.6 | | 5.0 | 14.3 | |
| Female | 12.2 | 46.2 | | 26.6 | 71.4 | | 3.0 | 16.7 | |
| Smoking history | | | 0.480 | | | 0.568 | | | 0.870 |
| Nonsmoker | 12.6 | 40.0 | | 37.8 | 66.7 | | 11.4 | 11.1 | |
| Smoker | 8.9 | 45.8 | | 26.3 | 52.6 | | 5.0 | 18.2 | |
| Asbestos exposure | | | 0.124 | | | 0.272 | | | 0.007 |
| Yes | 15.3 | 60.0 | | NR | 60.0 | | 12.6 | 60.0 | |
| No/unknown | 12.2 | 35.9 | | 26.3 | 58.3 | | 4.2 | 0.0 | |
| Location of tumor | | | 0.792 | | | 0.625 | | | 0.298 |
| Right | 12.6 | 45.2 | | 26.6 | 63.2 | | 2.9 | 16.7 | |
| Left | 15.3 | 39.1 | | NR | 53.3 | | 1.4 | 12.5 | |
| Bilateral pleural plaque | | | 0.744 | | | 0.479 | | | 0.746 |
| Yes | 12.6 | 60.0 | | NR | 80.0 | | 7.1 | 40.0 | |
| No | 17.2 | 38.6 | | 26.6 | 55.2 | | 5.0 | 6.7 | |
| Histologic subtype | | | <0.001 | | | | | | |
| Epithelioid | 26.6 | 58.8 | | | | | | | |
| Non-epithelioid | 5.0 | 15.0 | | | | | | | |
| Stage | | | 0.001 | | | 0.013 | | | 0.028 |
| I–II | NR | 70.0 | | NR | 85.7 | | 15.3 | 33.3 | |
| III–IV | 7.9 | 26.5 | | 17.2 | 40.0 | | 3.1 | 7.1 | |
| SUVmax* | | | 0.002 | | | 0.007 | | | 0.266 |
| > cutoff | 7.9 | 24.0 | | 12.2 | 42.9 | | 4.2 | 7.7 | |
| ≤ cutoff | 26.6 | 58.6 | | NR | 70.0 | | 11.4 | 28.6 | |
| EPP | | | 0.816 | | | 0.437 | | | 0.646 |
| Yes | 8.5 | 40.0 | | NR | 66.7 | | 4.4 | 0.0 | |
| No | 12.6 | 43.2 | | 26.3 | 57.1 | | 5.0 | 18.8 | |
| Chemotherapy | | | 0.031 | | | 0.931 | | | 0.005 |
| Pemetrexed/platinum | 22.5 | 47.2 | | 26.6 | 60.0 | | 8.5 | 18.2 | |
| None | 4.4 | 33.3 | | 37.8 | 55.6 | | 3.0 | 11.1 | |

*SUVmax cutoff; Total = 10.1, Epithelioid = 8.5, Non-epithelioid = 10.3

NR, not reached; SUVmax, maximum standardized uptake value; EPP, extra-pleural pneumonectomy

platinum-based chemotherapy were prognostic factors in the univariate analysis in this study, which were consistent with previous studies [5, 10, 11], and were evaluated in the multivariate analysis.

Previous studies compared SUVmax between MPM patients with epithelioid and non-epithelioid subtypes. Kadota et al. showed that pleomorphic subtype of epithelioid histology showed higher SUVmax than epithelioid non-pleomorphic subtype and was similar to non-epithelioid histology [14]. However, two studies reported no statistically significant differences in SUVmax between epithelioid and non-epithelioid subtypes in patients with MPM [16, 19].

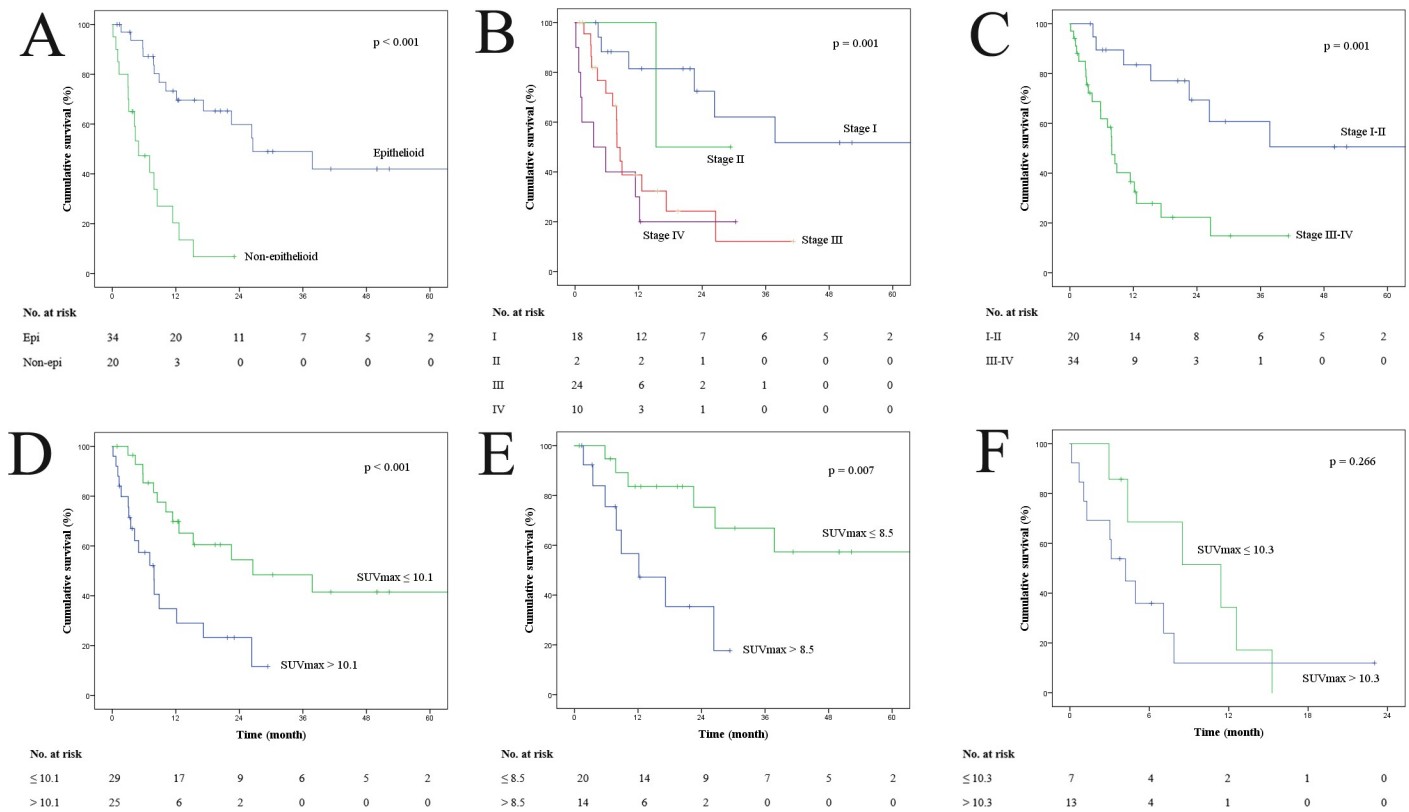

**Fig 2.** (A) Kaplan–Meier overall survival (OS) curve of all patients according to histologic subtype. (B, C) Kaplan–Meier OS curve of all patients according to stage. (D) Kaplan–Meier OS curve of all patients according to cutoff value of SUVmax. (E) Kaplan–Meier OS curve of patients with epithelioid subtype according to cutoff value of SUVmax. (F) Kaplan–Meier OS curve of patients with non-epithelioid subtype according to cutoff value of SUVmax.

And these studies have a limited number of patients with sarcomatoid subtype. In the present study, OS and SUVmax were significantly different between MPM patients with epithelioid subtype and those with non-epithelioid subtype. Furthermore, SUVmax was significantly higher in stage III–IV than in stage I–II.

In the present study, the cutoff value of SUVmax for death was 10.1 for all patients. However, the cutoff value of SUVmax was lower in patients with epithelioid subtype than in those with non-epithelioid subtype. In patients with non-epithelioid subtype, SUVmax was not associated with prognosis. Therefore, the cutoff value of SUVmax should be carefully interpreted with respect to tumor subtype. In previous studies, the cutoff values for SUVmax to

**Table 4. Multivariate analysis for overall survival.**

| | Total (n = 54) | | Epithelioid (n = 34) | | Non-epithelioid (n = 20) | |
|---|---|---|---|---|---|---|
| | Hazard ratio (95% CI) | P | Hazard ratio (95% CI) | P | Hazard ratio (95% CI) | P |
| SUVmax* (> cutoff vs. ≤ cutoff#) | 3.77 (1.58–9.01) | 0.003 | 5.65 (1.45–21.98) | 0.012 | 2.83 (0.79–10.1) | 0.111 |
| Histologic subtype (Epithelioid vs. non-epithelioid#) | 0.25 (0.10–0.64) | 0.003 | | | | |
| Stage (I–II vs. III–IV#) | 0.20 (0.08–0.52) | 0.001 | 0.15 (0.03–0.68) | 0.014 | 0.31 (0.06–1.61) | 0.163 |
| Chemotherapy (Pemetrexed/platinum vs. None#) | 0.34 (0.14–0.81) | 0.015 | 0.29 (0.06–1.47) | 0.134 | 0.28 (0.08–0.97) | 0.044 |

*SUVmax cutoff; Total = 10.1, Epithelioid = 8.5, Non-epithelioid = 10.3
#Reference

discriminate prognosis varied from 6 to 10 [12, 14, 16–18]. To serve as a prognostic factor in the clinical setting, a standardized method to determine the optimal cutoff value of SUVmax should be established.

There have been few biological explanations with respect to the relationship between SUVmax and survival in MPM patients. A previous study suggested that $^{18}$F-FDG uptake in MPM is influenced by glucose metabolism, phosphorylation of glucose, hypoxia, angiogenesis, cell proliferation (Ki-67), cell cycle regulators, and the mTOR pathway [22]. In addition, a positive correlation between mitotic count and SUVmax was reported in another study [14]. Further studies are needed to provide a biological explanation for the impact of SUVmax as a prognostic factor in MPM.

There are several staging systems available to demonstrate the prognostic significance of tumor stage on the survival of MPM patients. The eighth edition of the UICC/AJCC staging system for MPM has recently been developed [20]. Previous studies have reported that advanced AJCC clinical stage is associated with poor prognosis in MPM [23, 24], and the present study showed similar results. In addition, subgroup analysis showed that advanced stage was associated with poor prognosis in epithelioid subtype, but not in non-epithelioid subtype. The underlying cause of these results is unclear, but the non-epithelioid type may be associated with poor prognosis; moreover, the survival period is very short, even in early stages. Therefore, it is necessary to consider the histologic subtype when using clinical stage to predict prognosis in MPM patients; this should be confirmed by a prospective study in the future. Chemotherapy based on pemetrexed/platinum has been shown to improve survival in MPM patients [5, 24, 25]. In the present study, pemetrexed/platinum was administered to all patients receiving chemotherapy, and the survival rate was significantly improved, as in previous studies.

The present study had several limitations. First, relatively small sample size and limited number of events may invalidate the stability of the multivariable regression model in this study. The generalization of our results might potentially be limited by its retrospective nature and single-institution population. We also performed propensity score adjustment for histology subtype, stage and chemotherapy to validate the prognostic significance of SUVmax instead of the multivariable regression analysis. The hazard ratios (95% confidence interval) for total, epithelioid, and non-epithelioid histology were 1.83 (0.86–3.87; P = 0.114), 2.53 (0.83–7.67; P = 0.101), and 1.84 (0.59–5.75; P = 0.295), respectively. The hazard ratios of SUVmax after propensity score adjustment showed similar trends with the multivariable regression model but all the results from the propensity score adjustment were not statistically significant. Therefore, the results of current study from the multivariable model should be interpreted conservatively. Although there was no association between SUVmax and overall survival in non-epithelioid histology, a further prospective study using the multivariable model or propensity score adjustment is needed for the larger population in the future to elucidate the association between SUVmax and prognosis in epithelioid and non-epithelioid histology. Second, the histologic subtypes of the study subjects were not specifically defined in seven subjects who also underwent surgery. Finally, there were insufficient data on exposure to asbestos in 19 patients (35.2%).

In conclusion, the SUVmax on PET/CT is an independent prognostic factor in patients with MPM, especially in those with epithelioid subtype. The histologic subtype of MPM should be considered in evaluating the prognostic significance of SUVmax.

## Author Contributions

**Conceptualization:** Sang-Won Um.

**Data curation:** Jun Hyeok Lim.

**Formal analysis:** Jun Hyeok Lim, Sang-Won Um.

**Investigation:** Jun Hyeok Lim, Joon Young Choi, Yunjoo Im.

**Methodology:** Joon Young Choi.

**Project administration:** Sang-Won Um.

**Resources:** Joon Young Choi, Sang-Won Um.

**Supervision:** Sang-Won Um.

**Validation:** Yunjoo Im, Hongseok Yoo, Byung Woo Jhun, Byeong-Ho Jeong, Hye Yun Park, Kyungjong Lee, Hojoong Kim, O Jung Kwon, Joungho Han, Myung-Ju Ahn, Jhingook Kim.

**Writing – original draft:** Jun Hyeok Lim, Joon Young Choi, Sang-Won Um.

**Writing – review & editing:** Jun Hyeok Lim, Joon Young Choi, Yunjoo Im, Hongseok Yoo, Byung Woo Jhun, Byeong-Ho Jeong, Hye Yun Park, Kyungjong Lee, Hojoong Kim, O Jung Kwon, Joungho Han, Myung-Ju Ahn, Jhingook Kim, Sang-Won Um.

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
