## [Decision Letter · Decision Letter 0]

11 Nov 2019

PONE-D-19-27290

Prognostic value of SUVmax on 18F-fluorodeoxyglucose PET/CT scan in patients with malignant pleural mesothelioma

PLOS ONE

Dear Dr. Um,

Thank you for submitting your manuscript to PLOS ONE. After careful consideration, we feel that it has merit but does not fully meet PLOS ONE’s publication criteria as it currently stands. Therefore, we invite you to submit a revised version of the manuscript that addresses the points raised during the review process.

There are fairly few but important considerations expressed by the reviewers that will help improve this manuscript. Overall, it is a well-written paper describing a well-designed study. It is small and this limits some of the conclusions that can be drawn but it is interesting and merits publication.

Please revise taking into account the recommended revisions. With regard to Reviewer 2's concerns about the references for surgical candidacy, consider citing papers that actually discuss these issues (the ones you cite only describe the two operations not the evaluation for them and/or which to perform). One such paper is Wolf, Flores, Thorac Surg Clin. 2016 Aug;26(3):359-75.

The major issue with this study is that it is severely underpowered to detect differences and the reliability of the model is low as demonstrated by the large confidence intervals. This likely reflects overfitting with too many covariates and too few events/small sample size. I would consider redoing the analysis using propensity scores instead of a multivariable regression with so many covariates and only 54 patients.

Please also expand the limitations section as clearly there are additional limitations (some of which are described by the reviewers, but others that exist, including the single-institution, small sample size, short follow-up, limited number events that may invalidate the stability of the multivariable model, missing data with regard to asbestos exposure,among others) that should be described. Moreover, there should be some discussion about how the limitations might impact the results or why they are not as relevant as one might expect.

We would appreciate receiving your revised manuscript by November 30, 2019. To enhance the reproducibility of your results, we recommend that if applicable you deposit your laboratory protocols in protocols.io, where a protocol can be assigned its own identifier (DOI) such that it can be cited independently in the future. For instructions see: http://journals.plos.org/plosone/s/submission-guidelines#loc-laboratory-protocols

We look forward to receiving your revised manuscript.

Kind regards,

Andrea S. Wolf, M.D.

Academic Editor

PLOS ONE

Journal Requirements:

1. In the ethics statement in the manuscript and in the online submission form, please provide additional information about the patient records used in your retrospective study, including: a) whether all data were fully anonymized before you accessed them; b) the date range (month and year) during which patients' medical records were accessed.

Reviewers' comments:

Reviewer's Responses to Questions

**Comments to the Author**

1. Is the manuscript technically sound, and do the data support the conclusions?

Reviewer #1: Yes

Reviewer #2: Partly

2. Has the statistical analysis been performed appropriately and rigorously? 

Reviewer #1: Yes

Reviewer #2: Yes

3. Have the authors made all data underlying the findings in their manuscript fully available?

Reviewer #1: Yes

Reviewer #2: Yes

4. Is the manuscript presented in an intelligible fashion and written in standard English?

Reviewer #1: Yes

Reviewer #2: No

5. Review Comments to the Author

Reviewer #1: Please comment on how many patients had undergone talc or chemical pleurodesis? Pleurodesis can affect the SUV max. Please provide the number and how was analysis done accounting for those with and without pleurodesis .

Other than that the paper is well written and the results and discussion are well written

Reviewer #2: Dr. Um and colleagues have retrospectively evaluated the prognostic value of SUVmax on PET scans. This is an interesting exercise, especially to help differentiate within histologic subtypes.

1) The Zellos reference is outdated as it preceded the use of pemetrexed based therapy. Additionally, the survival referenced for multimodality therapy is not accurate.

2) The statement that no other agents have proven effective to treat mesothelioma is not accurate. Several other agents, including checkpoint inhibitors, vinorelbine and gemcitabine are active in mesothelioma.

3) It is inconsistent to state that trials of immunotherapies are underway and then cite references of completed and published clinical trials. Additionally, what does it mean that optimal candidates need to be selected and in what way do these references address that?

4) I am troubled by the suggestion that there may not be a relationship between SUV and prognosis in non-epithelioid histology. With so few patients, not finding an association does not provide meaningful data that a relationship does not exist, especially when other studies have demonstrated different results.

5) The discussion of Klabatsa and Lee is unclear. Did those studies account for subtypes within epithelioid histology. Please clarify the contrast between the two studies and Kadota study.

6) The use of terminology such as cutoff is unclear. For example, when stating that the cutoff value for death was 10.1, what does that mean? Did people below this level not die? Or is that a level at which the risk of death changes substantially? Or is it in reference to death within a certain time frame? Such comments must be clarified throughout the manuscript.

7) End of 1st paragraph says precious instead of previous.

6. PLOS authors have the option to publish the peer review history of their article (what does this mean?). If published, this will include your full peer review and any attached files.

Reviewer #1: Yes: Ritu R Gill

Reviewer #2: No

---

## [Author Response · Author response to Decision Letter 0]

10 Jan 2020

Point-to-point responses to the editor and reviewers’ comments; 

Editor

There are fairly few but important considerations expressed by the reviewers that will help improve this manuscript. Overall, it is a well-written paper describing a well-designed study. It is small and this limits some of the conclusions that can be drawn but it is interesting and merits publication.

C1: Please revise taking into account the recommended revisions. With regard to Reviewer 2's concerns about the references for surgical candidacy, consider citing papers that actually discuss these issues (the ones you cite only describe the two operations not the evaluation for them and/or which to perform). One such paper is Wolf, Flores, Thorac Surg Clin. 2016 Aug;26(3):359-75.

R1: We have added the reference in the revised manuscript.

C2: The major issue with this study is that it is severely underpowered to detect differences and the reliability of the model is low as demonstrated by the large confidence intervals. This likely reflects overfitting with too many covariates and too few events/small sample size. I would consider redoing the analysis using propensity scores instead of a multivariable regression with so many covariates and only 54 patients.

R2: We performed propensity score adjustment for histology subtype, stage and chemotherapy to validate the prognostic significance of SUVmax instead of the multivariable regression analysis as the editor recommended. The hazard ratios (95% confidence interval) for total, epithelioid, and non-epithelioid histology were 1.83 (0.86-3.87; P=0.114), 2.53 (0.83-7.67; P=0.101), and 1.84 (0.59-5.75; P=0.295), respectively. The hazard ratios of SUVmax after propensity score adjustment showed similar trends with the multivariable regression analysis but all the results from the propensity score adjustment were not statistically significant.

We have added this point as a limitation of this study to the Discussion section as follows; 

“First, relatively small sample size and limited number of events may invalidate the stability of the multivariable regression model in this study. The generalization of our results might potentially be limited by its retrospective nature and single-institution population. We also performed propensity score adjustment for histology subtype, stage and chemotherapy to validate the prognostic significance of SUVmax instead of the multivariable regression analysis. The hazard ratios (95% confidence interval) for total, epithelioid, and non-epithelioid histology were 1.83 (0.86-3.87; P=0.114), 2.53 (0.83-7.67; P=0.101), and 1.84 (0.59-5.75; P=0.295), respectively. The hazard ratios of SUVmax after propensity score adjustment showed similar trends with the multivariable regression model but all the results from the propensity score adjustment were not statistically significant. Therefore, the results of current study from the multivariable model should be interpreted conservatively. Although there was no association between SUVmax and overall survival in non-epithelioid histology, a further prospective study using the multivariable model or propensity score adjustment is needed for the larger population in the future to elucidate the association between SUVmax and prognosis in epithelioid and non-epithelioid histology.”

C3: Please also expand the limitations section as clearly there are additional limitations (some of which are described by the reviewers, but others that exist, including the single-institution, small sample size, short follow-up, limited number events that may invalidate the stability of the multivariable model, missing data with regard to asbestos exposure, among others) that should be described. Moreover, there should be some discussion about how the limitations might impact the results or why they are not as relevant as one might expect.

R3: We have modified the limitations of the study in the Discussion section as follows;

Original: “The present study had several limitations. First, the results of the study should be interpreted conservatively due to its retrospective nature and relatively small sample size. Second, the histologic subtypes of the study subjects were not specifically defined in seven subjects who also underwent surgery.” 

Revised: “The present study had several limitations. First, relatively small sample size and limited number of events may invalidate the stability of the multivariable regression model in this study. The generalization of our results might potentially be limited by its retrospective nature and single-institution population. We also performed propensity score adjustment for histology subtype, stage and chemotherapy to validate the prognostic significance of SUVmax instead of the multivariable regression analysis. The hazard ratios (95% confidence interval) for total, epithelioid, and non-epithelioid histology were 1.83 (0.86-3.87; P=0.114), 2.53 (0.83-7.67; P=0.101), and 1.84 (0.59-5.75; P=0.295), respectively. The hazard ratios of SUVmax after propensity score adjustment showed similar trends with the multivariable regression model but all the results from the propensity score adjustment were not statistically significant. Therefore, the results of current study from the multivariable model should be interpreted conservatively. Although there was no association between SUVmax and overall survival in non-epithelioid histology, a further prospective study using the multivariable model or propensity score adjustment is needed for the larger population in the future to elucidate the association between SUVmax and prognosis in epithelioid and non-epithelioid histology. Second, the histologic subtypes of the study subjects were not specifically defined in seven subjects who also underwent surgery. Finally, there were insufficient data on exposure to asbestos in 19 patients (35.2%).”

Reviewer #1

C4: Please comment on how many patients had undergone talc or chemical pleurodesis? Pleurodesis can affect the SUV max. Please provide the number and how was analysis done accounting for those with and without pleurodesis. Other than that the paper is well written and the results and discussion are well written

R4: Five patients had undergone talc or chemical pleurodesis before PET/CT scan. All 5 patients had epithelioid subtype of MPM. There was no statistically significant differences in the SUVmax between 5 patients who underwent pleurodesis (5.1 [3.5-14.7]) and 49 patients who did not undergo pleurodesis (10.0 [4.5-13.5]; P=0.662).

Reviewer #2

Dr. Um and colleagues have retrospectively evaluated the prognostic value of SUVmax on PET scans. This is an interesting exercise, especially to help differentiate within histologic subtypes.

C5: 1) The Zellos reference is outdated as it preceded the use of pemetrexed based therapy. Additionally, the survival referenced for multimodality therapy is not accurate. 2) The statement that no other agents have proven effective to treat mesothelioma is not accurate. Several other agents, including checkpoint inhibitors, vinorelbine and gemcitabine are active in mesothelioma. 3) It is inconsistent to state that trials of immunotherapies are underway and then cite references of completed and published clinical trials. Additionally, what does it mean that optimal candidates need to be selected and in what way do these references address that?

R5: Thank you so much for the comments. We have removed the Zellos reference and have modified the Introduction section as the reviewer #2 recommended:

Original: “Malignant pleural mesothelioma (MPM) is a rare but aggressive tumor that arises from pleural mesothelial cells. The prognosis of patients with MPM is poor, with median survival of 9–12 months despite multimodal therapy including surgery, chemotherapy, and radiotherapy [1]. Surgical methods (e.g., extra-pleural pneumonectomy [EPP] or pleurectomy/decortication) should be selected in accordance with the patient's condition [2, 3]. Among chemotherapeutic agents, a pemetrexed and platinum-based regimen has been recommended as a first-line agent because of its proven ability to improve the survival rate, but no other chemotherapeutic agents have been proven to effectively treat MPM [4]. In addition, clinical trials of new immunotherapeutic agents, such as pembrolizumab and nivolumab, are in progress, and it is important to select optimal candidates for these new agents [5-7].”

Revised: “Malignant pleural mesothelioma (MPM) is a rare but aggressive tumor that arises from pleural mesothelial cells. The prognosis of patients with MPM is poor, with a median survival of 20–29 months despite tri-modality treatment including surgery, chemotherapy, and radiotherapy [1, 2]. Surgical methods (e.g., extra-pleural pneumonectomy [EPP] or pleurectomy/decortication) should be selected in accordance with the patient's condition [3-4]. Among chemotherapeutic agents, a pemetrexed and platinum-based regimen has been recommended as a first-line treatment because of its proven ability to improve the survival rate [5, 6]. Immune checkpoint inhibitors, vinorelbine and gemcitabine are recommended as subsequent systemic therapy in the most recent guideline [6]. Pembrolizumab or nivolumab with (or without) ipilimumab showed promising results in recent clinical trials [7, 8, 9].”

References:

1. Krug LM, et al. Multicenter phase II trial of neoadjuvant pemetrexed plus cisplatin followed by extrapleural pneumonectomy and radiation for malignant pleural mesothelioma. J Clin Oncol 2009; 27:3007-3013.

2. Thieke C, et al. Long-term results in malignant pleural mesothelioma treated with neoadjuvant chemotherapy, extrapleural pneumonectomy and intensity-modulated radiotherapy. Radiat Oncol 2015: 10: 267.

3. Treasure T, et al. Extra-pleural pneumonectomy versus no extra-pleural pneumonectomy for patients with malignant pleural mesothelioma: clinical outcomes of the Mesothelioma and Radical Surgery (MARS) randomised feasibility study. The lancet oncology. 2011;12(8):763-72.

4. Wolf AS, Flores RM. Current treatment of mesothelioma: extrapleural pneumonectomy versus pleurectomy/decortication. Thoracic surgery clinics. 2016;26(3):359-75.

5. Vogelzang NJ, et al. Phase III study of pemetrexed in combination with cisplatin versus cisplatin alone in patients with malignant pleural mesothelioma. Journal of clinical oncology. 2003;21(14):2636-44.

6. Network NCC. NCCN malignant pleural mesothelioma guidelines, version 1.2020 Nov 27, 2019. Available from: https://www.nccn.org/professionals/physician_gls/pdf/mpm.pdf.

7. Scherpereel A, et al. Nivolumab or nivolumab plus ipilimumab in patients with relapsed malignant pleural mesothelioma (IFCT-1501 MAPS2): a multicentre, open-label, randomised, non-comparative, phase 2 trial. Lancet Oncol. 2019; 20(2): 239–253. 

8. Disselhorst MJ, et al. Ipilimumab and nivolumab in the treatment of recurrent malignant pleural mesothelioma (INITIATE): results of a prospective, single-arm, phase 2 trial. Lancet Respir Med. 2019; 7(3): 260– 270. 

9. Alley EW, et al. Clinical safety and activity of pembrolizumab in patients with malignant pleural mesothelioma (KEYNOTE-028): preliminary results from a non-randomised, open-label, phase 1b trial. Lancet Oncol. 2017; 18(5): 623–630

C6: 4) I am troubled by the suggestion that there may not be a relationship between SUV and prognosis in non-epithelioid histology. With so few patients, not finding an association does not provide meaningful data that a relationship does not exist, especially when other studies have demonstrated different results.

R6: We also agree with the reviewer’s opinion. We have added following sentences as a limitation of this study to the Discussion section.

“Although there was no association between SUVmax and overall survival in non-epithelioid histology, a further prospective study using the multivariable model or propensity score adjustment is needed for the larger population in the future to elucidate the association between SUVmax and prognosis in epithelioid and non-epithelioid histology.”

C7: 5) The discussion of Klabatsa and Lee is unclear. Did those studies account for subtypes within epithelioid histology. Please clarify the contrast between the two studies and Kadota study.

R7: In Kadota's study, pleomorphic subtype of epithelioid histology showed higher SUVmax than epithelioid non-pleomorphic subtype and was similar to non-epithelioid histology. However, in Klabatsa and Lee's studies, there was no significant difference in SUVmax between epithelioid and non-epithelioid subtypes. We have modified the manuscript to clarify the differences of previous studies as follows;

Original: “Kadota et al. showed that SUVmax in MPM with epithelioid nonpleomorphic subtype was lower than that in MPM with epithelioid pleomorphic subtype and that in MPM with non-epithelioid subtype [14]. In contrast, two studies reported no statistically significant differences in PET parameters between MPM patients with epithelioid and non-epithelioid subtypes [16, 19].”

Revised: “Kadota et al. showed that pleomorphic subtype of epithelioid histology showed higher SUVmax than epithelioid non-pleomorphic subtype and was similar to non-epithelioid histology [14]. However, two studies reported no statistically significant differences in SUVmax between epithelioid and non-epithelioid subtypes in patients with MPM [16, 19].”

C8: 6) The use of terminology such as cutoff is unclear. For example, when stating that the cutoff value for death was 10.1, what does that mean? Did people below this level not die? Or is that a level at which the risk of death changes substantially? Or is it in reference to death within a certain time frame? Such comments must be clarified throughout the manuscript.

R8: As the reviewer pointed out, we have clarified the meaning of the cutoff value in the Materials and Methods section (Statistical analysis) as follows; 

Original: “Receiver operating characteristic (ROC) curves of the SUVmax for the prediction of mortality were generated to determine the cutoff value that yielded optimal sensitivity and specificity.”

Revised: “Receiver operating characteristic (ROC) curves were plotted to determine the optimal cutoff values of SUVmax that yielded the maximal sensitivity plus specificity of predicting the overall survival. The patient population was subdivided using the cutoff values of SUVmax from the ROC curves, and the duration of overall survival was compared between groups.”

C9: 7) End of 1st paragraph says precious instead of previous.

R9: We have corrected the typo.

---

## [Editor Report · Decision Letter 1]

4 Feb 2020

Prognostic value of SUVmax on 18F-fluorodeoxyglucose PET/CT scan in patients with malignant pleural mesothelioma

PONE-D-19-27290R1

Dear Dr. Um,

We are pleased to inform you that your manuscript has been judged scientifically suitable for publication and will be formally accepted for publication once it complies with all outstanding technical requirements.

With kind regards,

Andrea S. Wolf, M.D.

Academic Editor

PLOS ONE

Additional Editor Comments (optional):

The authors have performed additional analysis and made substantial revisions that clarify the significance of this work. This manuscript should be published.
---

## [Editor Report · Acceptance letter]

6 Feb 2020

PONE-D-19-27290R1 

Prognostic value of SUVmax on 18F-fluorodeoxyglucose PET/CT scan in patients with malignant pleural mesothelioma 

Dear Dr. Um:

I am pleased to inform you that your manuscript has been deemed suitable for publication in PLOS ONE. Congratulations! Your manuscript is now with our production department. 

With kind regards,

on behalf of

Dr. Andrea S. Wolf 

Academic Editor

PLOS ONE